# Atypical Ductal Hyperplasia and Lobular In Situ Neoplasm: High-Risk Lesions Challenging Breast Cancer Prevention

**DOI:** 10.3390/cancers16040837

**Published:** 2024-02-19

**Authors:** Luca Nicosia, Luciano Mariano, Giuseppe Pellegrino, Federica Ferrari, Filippo Pesapane, Anna Carla Bozzini, Samuele Frassoni, Vincenzo Bagnardi, Davide Pupo, Giovanni Mazzarol, Elisa De Camilli, Claudia Sangalli, Massimo Venturini, Maria Pizzamiglio, Enrico Cassano

**Affiliations:** 1Breast Imaging Division, Radiology Department, (IEO) European Institute of Oncology IRCCS, 20141 Milan, Italy; luciano.mariano@ieo.it (L.M.); federica.ferrari@ieo.it (F.F.); filippo.pesapane@ieo.it (F.P.); anna.bozzini@ieo.it (A.C.B.); davide.pupo@ieo.it (D.P.); maria.pizzamiglio@ieo.it (M.P.); enrico.cassano@ieo.it (E.C.); 2Postgraduate School of Radiodiagnostics, University of Milan, 20122 Milan, Italy; giuseppe.pellegrino@unimi.it; 3Department of Statistics and Quantitative Methods, University of Milan-Bicocca, 20126 Milan, Italy; samuele.frassoni@unimib.it (S.F.); vincenzo.bagnardi@unimib.it (V.B.); 4Division of Pathology, (IEO) European Institute of Oncology IRCCS, 20141 Milan, Italy; giovanni.mazzarol@ieo.it (G.M.); elisa.decamilli@ieo.it (E.D.C.); 5Data Management, (IEO) European Institute of Oncology IRCCS, 20141 Milan, Italy; claudia.sangalli@ieo.it; 6Diagnostic and Interventional Radiology Unit, ASST Settelaghi, Insubria University, 21100 Varese, Italy; massimo.venturini@uninsubria.it

**Keywords:** high-risk breast lesions, atypical ductal hyperplasia, lobular in situ neoplasm, B3 lesion

## Abstract

**Simple Summary:**

This retrospective study investigates histopathological outcomes, upgrade rates, and disease-free survival (DFS) in high-risk breast lesions, including atypical ductal hyperplasia (ADH or DIN1b) and lobular in situ neoplasms (LIN 1 and 2), following Vacuum-Assisted Breast Biopsy (VABB) and surgical excision. Focusing on 320 patients who underwent stereotactic VABB, with 246 individuals diagnosed with ADH (120) or LIN (126), the study addresses the challenge posed by these lesions due to their association with synchronous or adjacent Breast Cancer (BC) and increased future BC risk. The study underscores the importance of a multidisciplinary approach, acknowledging the evolving role of VABB, and emphasizes the need for careful follow-up, particularly for lobular lesions. It offers valuable insights for clinicians navigating the complex landscape of high-risk breast lesions, advocating for heightened awareness and vigilance in managing these lesions and contributing to the ongoing refinement of clinical strategies in BC care.

**Abstract:**

This retrospective study investigates the histopathological outcomes, upgrade rates, and disease-free survival (DFS) of high-risk breast lesions, including atypical ductal hyperplasia (ADH or DIN1b) and lobular in situ neoplasms (LIN), following Vacuum-Assisted Breast Biopsy (VABB) and surgical excision. The study addresses the challenge posed by these lesions due to their association with synchronous or adjacent Breast Cancer (BC) and increased future BC risk. The research, comprising 320 patients who underwent stereotactic VABB, focuses on 246 individuals with a diagnosis of ADH (120) or LIN (126) observed at follow-up. Pathological assessments, categorized by the UK B-coding system, were conducted, and biopsy samples were compared with corresponding excision specimens to determine upgrade rates for in situ or invasive carcinoma. Surgical excision was consistently performed for diagnosed ADH or LIN. Finally, patient follow-ups were assessed and compared between LIN and ADH groups to identify recurrence signs, defined as histologically confirmed breast lesions on either the same or opposite side. The results reveal that 176 (71.5%) patients showed no upgrade post-surgery, with ADH exhibiting a higher upgrade rate to in situ pathology than LIN1 (Atypical Lobular Hyperplasia, ALH)/LIN2 (Low-Grade Lobular in situ Carcinoma, LCIS) (38% vs. 20%, respectively, *p*-value = 0.002). Considering only patients without upgrade, DFS at 10 years was 77%, 64%, and 72% for ADH, LIN1, and LIN2 patients, respectively (*p*-value = 0.92). The study underscores the importance of a multidisciplinary approach, recognizing the evolving role of VABB. It emphasizes the need for careful follow-up, particularly for lobular lesions, offering valuable insights for clinicians navigating the complex landscape of high-risk breast lesions. The findings advocate for heightened awareness and vigilance in managing these lesions, contributing to the ongoing refinement of clinical strategies in BC care.

## 1. Introduction

High-risk breast lesions are a group of heterogeneous cell proliferations that can be associated with synchronous or adjacent Breast Cancer (BC) and that confer an increased future risk of developing BC [1]. Atypical ductal hyperplasia (ADH-DIN1b) and lobular in situ neoplasms (LINs), including Atypical Lobular Hyperplasia (ALH-LIN1) and Low-Grade Lobular in situ Carcinoma (LIN 2), are among the most commonly diagnosed high-risk lesions diagnosed after a breast biopsy [2]. The frequency of their diagnosis has increased with the advent of Digital Mammography (DM) screening, with figures of 12–17% of cases diagnosed due to microcalcifications or architectural distortions [2].

Considering their still benign nature but their relatively high risk of upgrading to a malignant disease, these borderline lesions fall under the B3 category, according to the B-coding system, as “lesions of uncertain malignant potential” [3]. The estimated cumulative incidence of upgrade to BC of these lesions is around 30% at 25 years of follow-up, with a maximum in the first 5 years after the diagnosis [4,5] and with ADH having the highest upgrade rate to malignancy after excisional biopsy [6]. Indeed, the histopathological distinction between ADH and Ductal in situ Carcinoma (DCIS) is based on size/extent criteria, and it may be difficult to clearly outline a diagnosis, especially after examining limited samples obtained by a core biopsy (CB). Regarding LIN, sampling errors owing to the often-spreading nature of the disease also typically require surgical excision to obtain a satisfying sample to analyze, with a very variable upgrade rate according to recent reviews ranging from 2 to 40% [7,8].

For these reasons, the first approach to managing these findings has traditionally been surgical. However, in the last decade, the increased detection of B3 lesions has been associated with a drop in the positive predictive value of malignancy to approximately 10% [9]: this inevitably leads to numerous surgical resections of benign lesions, at the price of negative psychological and economic impacts on the patient and system. Although today, the general trend towards minimally invasive treatments of B3 lesions has highlighted the role of Vacuum-Assisted Breast Biopsy and Excision (VABB and VAE) with relatively low rates of upgrade [10], the literature shows a wide variability in results in terms of correct lesion characterization without surgical excision, especially regarding lesion size, with an overall orientation still tending towards the more conservative care of such lesions [11].

In light of the controversial management of these conditions, this study aims to assess and compare the histopathological results of ADH, LIN1, and LIN2 after the VABB procedure and surgical excision to evaluate the percentage rate of upgrade after surgery successively. Ten years of disease-free survival (DFS) for each of the three classes was also assessed. The ultimate goal of this study is to present our center’s experience, hoping to provide additional information to manage patients diagnosed with this tricky class of lesions appropriately and indicate the best possible plan specifically tailored to every patient’s disease and risk profile.

## 2. Materials and Methods

This retrospective research study adhered to ethical standards and received approval from the Institutional Review Board (IRB) with the identification number UID 2897, granted on 24 September 2021.

A retrospective analysis of patient data was conducted, specifically focusing on individuals diagnosed with ADH, LIN1, and LIN2 following stereotactic-guided VABB procedures. From the pathology anatomy reports, we selected all consecutive cases of ADH, LIN1, and LIN2 diagnosed between 1999 and 2016 (at least five years of follow-up), excluding mixed forms (e.g., ADH and LIN in the same sample).

From the pathologist’s report, information was obtained on the number of biopsy cores obtained for each patient and the presence of disease in the samples with and without microcalcifications (in the Institute, an X-ray of the biopsy material is taken to separate the samples with microcalcifications from those without microcalcifications).

All lesions were identified by DM screening, and VABB was performed using needle Gauges of 11 G, 10 G, or 8 G (in 82.5% of cases, biopsy was performed with an 11G needle). The histological assessment of the biopsy samples was categorized according to the UK B-coding system, classifying lesions as B1 to B5 [12,13]. When ADH or LIN (1 or 2) was diagnosed based on the biopsy specimen, the surgical excision of the affected breast tissue was always performed. Upon surgical excision, breast tissue specimens were processed by institutional guidelines. Each biopsy was individually compared with its corresponding excision specimen to assess the upgrade rate to BC, defined as the finding of in situ carcinoma (ISC-B5a) or invasive carcinoma (IC-B5b) in the surgical specimen. This upgrade rate was compared between LIN and ADH. Lastly, they were assessed and compared (LIN vs. ADH) in patient follow-ups to detect signs of recurrence, defined as histologically confirmed ipsilateral or contralateral breast lesions (classified as B3, B4, or B5) detected during periodic radiological examinations conducted after ADH or LIN surgery. We identified the percentage of patients receiving tamoxifen therapy during follow-up.

Pathological diagnosis was according to WHO immuno-morphological criteria (WHO Classification of Tumours, 5th Edition, 2019).

### Statistical Analysis

Continuous data were reported as medians and ranges. Categorical data were reported as counts and percentages. Wilcoxon’s signed-rank tests for continuous variables and Chi-squared tests (or Fisher’s exact tests, when appropriate) for binary variables were used to compare the distribution of the evaluated descriptive variables and the upgrade rates between ADH and LIN patients.

DFS was defined as the time from the date of surgery to subsequent recurrence (ipsilateral or contralateral), another primary tumor, death, or last contact, whichever occurred first. DFS was estimated using the Kaplan–Meier method, considering only patients without an upgrade. The log-rank test assessed differences between ADH, LIN1, and LIN2 patients.

A *p*-value less than 0.05 was considered statistically significant. All analyses were performed with the statistical software SAS 9.4 (SAS Institute, Cary, NC, USA).

## 3. Results

A total of 320 female patients undergoing stereotactic VABB were selected: 141 with ADH and 179 with LIN. In total, 74 patients (21 ADH and 53 LIN) were excluded due to a prior BC diagnosis. Therefore, 246 patients were enrolled (120 with ADH and 126 with LIN). Of these, 176 did not have an upgrade at the subsequent surgery and were included in the DFS analysis (a flowchart of the study is presented in Figure 1).

The median age at biopsy was 52 years (range: 31–78) among ADH patients and 49 years (range: 37–70) for patients with LIN (*p*-value = 0.056). The distribution of patients by age group is presented in Appendix A. The median lesion size on the radiological image at biopsy was 15 mm (range: 5–100) for ADH and 13 mm (range: 3–80) for LIN patients (*p*-value = 0.17). In all cases, VABB was performed. In our series, ADH always presented as microcalcifications on imaging; 19 out of 179 (10.6%) LIN cases presented as parenchymal distortion, while the remaining percentage was microcalcifications.

Considering only specimens with microcalcifications, the disease occurred in 47 patients with ADH (48.5%) and 19 patients with LIN (18.1%, *p*-value < 0.001).

These descriptive data are summarized in Table 1.

Among the 120 ADH patients, 45 (38%) showed an upgrade, while it was observed only in 25 (20%) LIN1/2 patients (*p*-value = 0.002). The upgrade rate to B5a was 29% and 6% among ADH and LIN1/2 patients, respectively (*p*-value < 0.001), while the upgrade rate to B5b was similar in the two groups (8% for ADH and 13% among LIN1/2 patients, *p*-value = 0.58). From our overall series, we obtained, at surgery, 43 upgraded cases to in situ BC and 27 upgraded cases to invasive BC. Regarding in situ BC underestimations, in 34 out of 43 cases, there was an underestimation of low-grade DCIS, and in 9 out of 43 an underestimation of intermediate DCIS.

The underestimation of the biopsy was easier when the disease was not confined to samples with microcalcifications. In all cases, the underestimated disease was focal and non-extensive.

The distribution and comparison of upgrade to surgery between ADH and LIN patients is reported in Table 2 and Table 3.

Table 3 shows that upgrades to in situ BC in our series are more frequent for ADH than LIN 1 and LIN2 (*p*-value < 0.001). There are no statistically significant differences in progression to invasive BC between ADH, LIN1, and LIN2.

For DFS evaluation, only patients without upgrades were considered (176 patients: 75 ADH, 33 LIN 1, and 68 LIN 2). The median time to follow-up was 6.3 years (Q1–Q3: 2.5–10.3). Twelve (16%) DFS events (one ipsilateral and the same quadrant, two ipsilateral and a different quadrant, nine contralateral) were observed among ADH patients. Ten (30%) events (three ipsilateral and the same quadrant, three ipsilateral and a different quadrant, two contralateral, and two tumors in organs other than the breast) and eighteen (26%) events (seven ipsilateral and the same quadrant, three ipsilateral and a different quadrant, six contralateral, and two tumors in organs other than the breast) were observed among LIN1 and LIN2 patients, respectively. In our case series, 47/173 (27.1%), tamoxifen therapy was performed. The oncologist introduced this type of therapy and considered the various risk factors after a multidisciplinary discussion. The treatment was proposed to those patients with more critical risk factors, such as grade I relatives with breast and ovarian neoplasms.

For ADH, we had 12 events during follow-up: 5 in situ carcinomas in the contralateral breast, 4 invasive carcinomas in the contralateral breast, and 1 ipsilateral invasive carcinoma. For LIN1, we had 10 events at follow-up: 6 ipsilateral invasive carcinomas, 3 ipsilateral in situ carcinomas, and 1 contralateral invasive carcinoma. For LIN 2, we had four ipsilateral invasive, three ipsilateral in situ, and one contralateral invasive.

The estimated 5-year DFS was 80% (95% CI: 65–89%) for ADH, 84% (95% CI: 65–93%) for LIN1, and 88% (95% CI: 77–94%) for LIN2. The 10-year DFS was 77% (95% CI: 62–87%) for ADH, 64% (95% CI: 42–79%) for LIN1, and 72% (95% CI: 58–83%) for LIN2 (*p*-value = 0.92, Figure 2).

## 4. Discussion

ADH and LIN are a heterogeneous group of breast lesions with a non-negligible risk of future malignancy, posing an ongoing challenge for breast physicians. Categorized as “B3” within the B-coding system due to uncertain malignant potential [14], their diagnosis has improved in recent years thanks to increasingly efficient DM screening programs [15]. While evidence suggests that these lesions may be nonobligate precursor lesions [16], they are generally managed as risk indicators rather than precursor lesions, as not all patients will develop BC. The BC that does develop subsequently may occur in either breast and not necessarily at the site of the atypia. Managing B3 lesions has emerged as a prominent topic in BC imaging, prompting extensive discussion in several publications [14,17].

The current study highlights the varying degree of diagnostic detection between ADH and LIN following the stereotactic VABB procedure. Additionally, it compares patient outcomes with diagnoses of ADH, LIN1, and LIN2, assessing their upgrade rate at surgery.

From our results of samples with only microcalcifications, retrieved by the VABB technique, comes to light a comparatively greater likelihood of detecting disease in ADH than in LIN (48.5% vs. 18.1%), in relation to the disparities concerning incidence and cytoarchitectural differences. ADH is indeed one of the most common B3 lesions diagnosed by stereotactic VABB [18], representing a clear majority (81.6%) of these found as microcalcification on DM [19]. It is characterized by an intraductal clonal proliferation of uniformly spaced monotonous cells with an atypical architecture, resulting in necrosis and calcium salt deposits (Figure 3a–d) [20]. In contrast, LIN is usually an incidental finding on breast biopsies performed for other reasons [6,21], lacking a typical imaging pattern, histologically defined by the neoplastic proliferation of small dyscohesive epithelial cells, filling the involved lobules or being able to spread along the ducts, and interposing between myoepithelium and secretory epithelium (Pagetoid spread) (Figure 3f–i).

According to the new International Consensus Conference [18], the upgrade rate of B3 lesions at post-surgical histological evaluation ranges from 7.3 to 57% for ADH and 4 to 67% for LIN, with significant variations among different diagnostic procedures and types of needles employed. Notably, smaller samples obtained by CB resulted in higher upgrade events [22], correlating their prevalence more to the sampling technique than the procedure’s efficacy. Rageth et al. [19] compared the pathological findings of 207 patients diagnosed with ADH (57 by CB and 151 by VABB) with surgical pieces, correlating the underestimation rate with the choice of CB (57% upgrade rate vs. 33% for VABB), the multifocality of disease, and the absence of calcification in the retrieved samples. Specifically analyzing our data, we reported an upgrade rate to malignancy at surgery for ADH and LIN of 38% and 20%, respectively. This result is consistent with literature reports that reflect a higher upgrade rate for ADH, both in relation to its accessible and more frequent DM detection [19], but especially for its pathological similarities to DCIS [23,24]. WHO defines ADH as a low-grade DCIS of limited extent (partial filling of the involved duct or completely filling the duct(s) but <2 mm [25] or <2 ducts involved [24]). Since the true ADH extent cannot be adequately evaluated in minimally invasive biopsy specimens, most of the experts still recommended a surgical approach for ADH lesions after diagnosis [18]. However, if the target lesion has been entirely excised by VABB or VAE, observation and DM follow-up are also suggested [18]. In a systematic review by Schiaffino et al. [26], out of 6458 ADH cases analyzed (5911 managed with surgical excision and 547 with follow-up), lesions undergoing observation showed a lower upgrade rate than those treated surgically (5% vs. 29%). These conservatively controlled lesions were generally smaller, fully treated after diagnostic excision, or diagnosed in low-risk women. However, achieving an adequate negative predictive value is still insufficient, so surgical excision for ADH remains the safest option.

The management of LIN lesions is also controversial, primarily due to their atypical radiological or clinical presentation [27] and their limited tendency for growth [28]. Establishing a radio-pathological correlation is essential for defining appropriate follow-up protocols. Research indicates that a radiologic–pathologically concordant LIN diagnosed on an excisional biopsy by VABB, preferably VAE, no longer requires surgical excision, provided excision is not indicated for the targeted lesion [29,30,31]. Nevertheless, considering the broad range of factors to be analyzed for each specific case, a thorough evaluation performed by a multidisciplinary team, especially on these borderline B3 lesions, is mandatory to tailor the treatment or follow-up plan appropriately to each lesion and each patient.

Our research also assessed DFS for the three lesions to understand patient management strategies better. Only patients without an upgrade at surgery (176) were considered to assess DFS, comprising 75 with ADH, 33 with LIN1, and 68 with LIN2. There were no statistically significant differences in DFS rates between ADH (5-year DFS: 80%, 10-year DFS: 77%), LIN1 (5-year DFS: 84%, 10-year DFS: 64%), and LIN2 (5-year DFS: 88%, 10-year DFS: 72%) patients (*p*-value = 0.92), suggesting comparable DFS outcomes across the three groups. Nevertheless, the overall rate of developing subsequent BC during follow-up after the diagnosis and excision of our B3 lesions was 23%. This result provides evidence that women with these lesions should be defined as “high-risk”. Recent studies [21,26] evaluating age-specific 10-year absolute risk to realize risk-stratified BC screening indicated a threshold of 6% to define “high-risk” women. In order, the three histotypes with a higher risk of future cancer were LIN1 (30%), LIN2 (26%), and ADH (16%); thus, in our opinion and according to the literature [32,33], these three categories could benefit from a tailored approach to surveillance, and patients with these lesions should not be discharged from clinical and radiological follow-up [34,35]. Recent guidelines published by several breast societies suggest that high-risk women should undergo an annual DM examination with an additional breast MRI or breast US or contrast-enhanced mammography when MRI is contraindicated or unavailable [36,37,38]. Indeed, according to our results, women with B3 lesions should be advised to follow this pathway. Considering that the diagnosis of BC occurs, on average, 38 months after the initial diagnosis of a B3 lesion, we suggest that undergoing at least a 5-year follow-up, with annual bilateral DM and semi-annual breast US examinations, represents a reasonable and cost-effective option for these patients.

The study’s main limitation is its retrospective nature. Furthermore, the analysis of DFS does not consider many factors that could influence it, such as comorbidities and tamoxifen therapy.

## 5. Conclusions

The management of B3 breast lesions can be considered one of the most debated topics, prompting multidisciplinary meetings to evaluate the best approach with these patients. According to data in the literature, ADH most commonly indicates operative management, while LIN1 and 2 are based on the radiopathologic degree of correlation, lesion, and patient characteristics. The increasing role of VAE could help manage these lesions, especially when complete excision is technically possible. The DFS data reported in this article confirm that a clinical and radiological follow-up with a yearly bilateral DM and US breast scan should be proposed for at least 5 years. This approach may translate to the earlier diagnosis of BC among patients with previous B3 lesion excision, with clear clinical benefits for these patients.

## Figures and Tables

**Figure 1 cancers-16-00837-f001:**
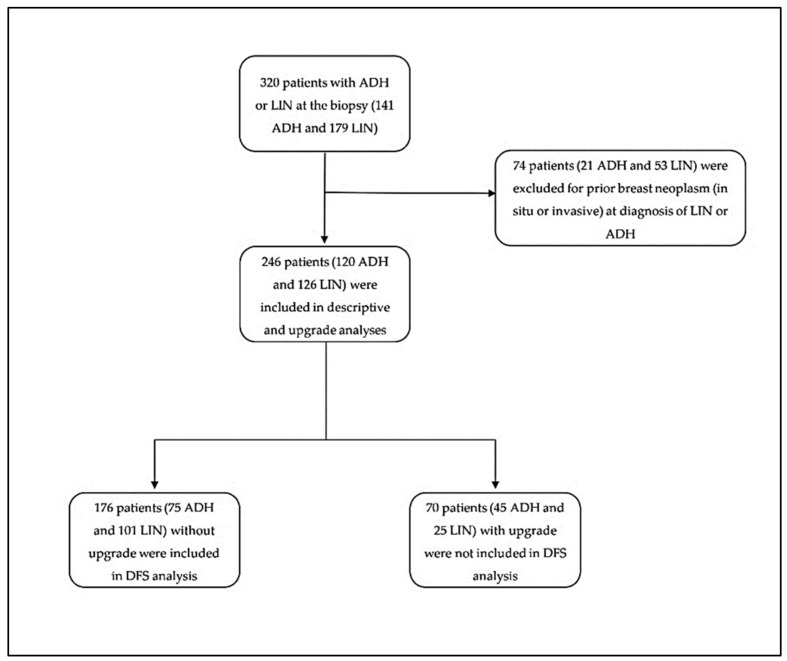
Flowchart diagram of the inclusion and exclusion criteria of the study.

**Figure 2 cancers-16-00837-f002:**
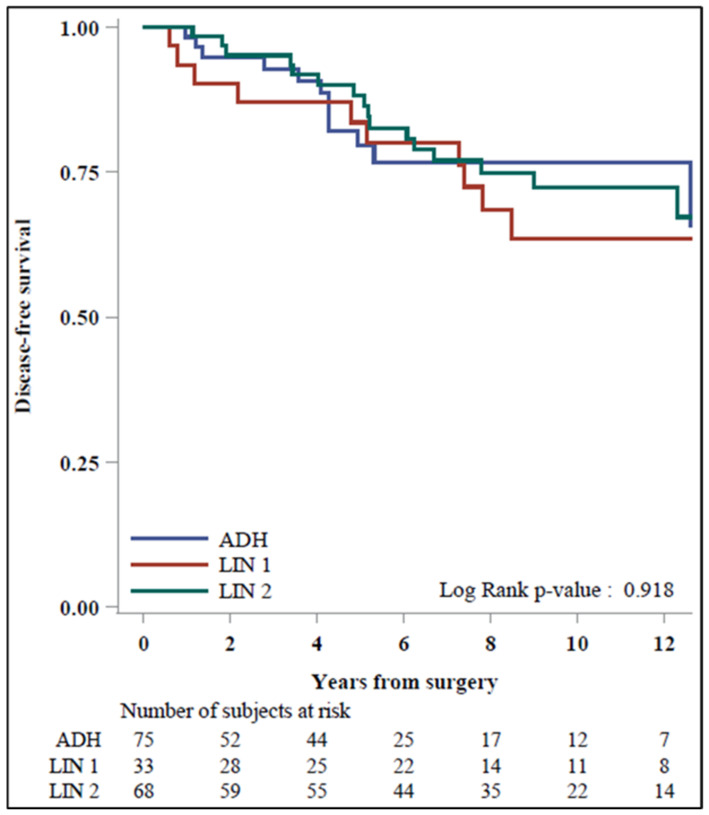
Disease-free survival among ADH and LIN patients (*n* = 176).

**Figure 3 cancers-16-00837-f003:**
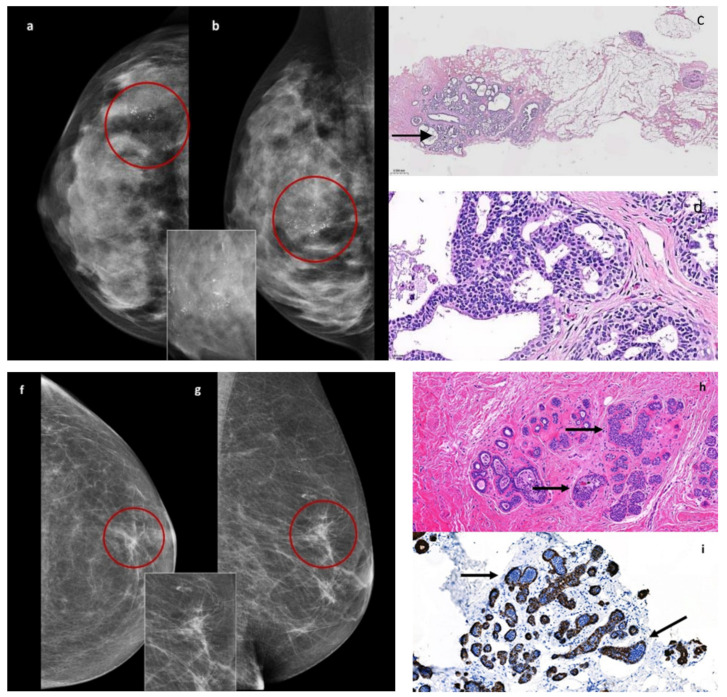
Atypical ductal hyperplasia (ADH). Unilateral craniocaudal (**a**) and mediolateral (**b**) mammograms of the right breast show amorphous microcalcifications with segmental distribution (rim) in the outer quadrants. The inset shows a higher magnification of amorphous microcalcifications. A stereotactic Vacuum-Assisted Breast Biopsy-diagnosed ADH. Hematoxy0lin and eosin stain (**c**) of a histological section of DIN1b (2×) with central microcalcification (arrow) adjacent to normal ducts on the right. High-power magnification clearly depicts low-grade cytoarchitectural atypia (**d**). Low-Grade Lobular in situ Carcinoma (LIN2). Unilateral craniocaudal (**f**) and mediolateral (**g**) mammograms of the left breast show an architectural distortion with fine microcalcifications associated (rim) in the upper para-areolar area. The inset shows a higher magnification of parenchymal distortion. A stereotactic Vacuum-Assisted Breast Biopsy-diagnosed LIN2. Hematoxylin and eosin stain (**h**) of a histological section of LIN (20×) with a lobular proliferation of low-grade epithelial cells confined to less than 50% of the tubule–lobular unit (arrows). Immunohistochemical stain of LIN with E-cadherin immunoreactivity (**i**) depicts negative (arrows) LIN cells intermixed with ductal positive cells.

**Table 1 cancers-16-00837-t001:** Descriptive variables among ADH and LIN patients (*n* = 246).

Variable	Level	ADH (*n* = 120)	LIN 1/2 (*n* = 126)	*p*-Value
**Age at VABB**, median (min–max)		52 (31–78)	49 (37–70)	0.056
Dimension of lesion, median (min–max)		15 (5–100)	13 (3–80)	0.17
Missing		0	21	
**Disease-only specimen with micro**, *n* (%)	No	50 (51.5)	86 (81.9)	<0.001
	Yes	47 (48.5)	19 (18.1)	
	Missing	23	21	

**Table 2 cancers-16-00837-t002:** Percentage of upgrade to surgery (ADH, LIN1, and LIN2).

At Surgery	ADH	LIN 1/2	LIN 1	LIN 2
	*n* (% col)			
**No Upgrade** (B2 or B3)	75 (63)	101 (80)	33 (83)	68 (79)
B2	*36 (30)*	*32 (25)*	*15 (38)*	*17 (20)*
B3	*39 (33)*	*69 (55)*	*18 (45)*	*51 (59)*
**Upgrade** (B5a or B5b)	45 (38)	25 (20)	7 (18)	18 (21)
B5a	*35 (29)*	*8 (6)*	*2 (5)*	*6 (7)*
B5b	*10 (8)*	*17 (13)*	*5 (13)*	*12 (14)*
**Total**	120	126	40	86

**Table 3 cancers-16-00837-t003:** Comparison of the upgrade rate to surgery (ADH, LIN1, and LIN2).

		*p*-Value
**Upgrade vs. No Upgrade**	ADH vs. LIN 1/2	0.002
	ADH vs. LIN 1	0.019
	ADH vs. LIN 2	0.011
**B5a vs. No Upgrade**	ADH vs. LIN 1/2	<0.001
	ADH vs. LIN 1	0.002
	ADH vs. LIN 2	<0.001
**B5b vs. No Upgrade**	ADH vs. LIN 1/2	0.58
	ADH vs. LIN 1	0.78
	ADH vs. LIN 2	0.54

## Data Availability

The data presented in this study are available on request from the corresponding author. The data are not publicly available due to privacy concerns, in accordance with GDPR.

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
