# Peer review of "Atypical Ductal Hyperplasia and Lobular In Situ Neoplasm: High-Risk Lesions Challenging Breast Cancer Prevention"

_cancers, 2024, doi:10.3390/cancers16040837_

Round 1

Reviewer 1 Report

Comments and Suggestions for Authors

In the article “Unveiling Challenges in High-risk Breast Lesions: Atypical Ductal Hyperplasia and Lobular in situ Neoplasm” the authors set out to evaluate upgrade rates of the low-grade atypical in situ proliferations detected on vacuum assisted breast biopsies. Overall, I think this article does have interesting data but it needs to be further discussed/explained and some key data is missing. The article would benefit from an English Language review as much of it is confusing or structured in an odd way.

-        The title doesn’t accurately reflect the content of the article. This article is focused on vacuum assisted breast biopsies and their subsequent upgrade rate. Also, the title doesn’t indicate what the challenge is. Unveiling is a bold term to use.

-        Throughout the article the terms “upstage” or “upstaging” is used when I think it should all be “upgrade.” However, I speak American English so this may be used in other English-speaking countries.

-        As someone from the US who practices breast pathology, I am not as familiar with the UK B-coding system or the DIN and LIN categories. It may be worth expanding on these for those who may not be as familiar. Or include those categories with ones that others may be familiar with as I think it would reach a greater audience.

-        Materials and Methods. How were the cases retrospectively identified? By the pathology or imaging? What timeframe were they collected? Were they consecutive cases? Further expand on the details of the collection.

-        Results.

o   ADH and ALH/LCIS are often seen together in specimens. Were there no biopsies with both or were they excluded?

o   I assume these were all women but it is not explicitly stated.

o   Line 142. What do you mean by size? The size of the atypia? As noted, linear extent is often used to distinguish ADH from low-grade DCIS. If you have a 15 mm focus of ADH then one would likely call that DCIS. Or was this dimension the imaging target?

o   At our institution vacuum assist is most commonly used in cases of calcifications. In this study not all cases had calcifications. What was the reason for vacuum assist? And were all of these calcifications identified on specimen radiograph, the histology, or both?

o   What were the calcifications associated with in the histologic sections?

o   Gauge of the needle was recorded as noted in the methods. An analysis evaluating needle gauge and upgrade rate would be good to include.

o   Table 1. What is MMT? It isn’t defined anywhere. Include the definition of this abbreviation in the caption. “Disease only specimen with micro” is confusing and I’m not sure what you mean by this. Based on the text I assume this is referring to the presence or absence of calcifications. Please reword. I would also note in the text that some cases had missing data about calcifications.

o   I think the formatting of Table 2 has been altered. I don’t know how to read it the way it currently is.

o   What is Table 3 demonstrating? There is no great explanation in the text.

o   Lines 168-169. Use ipsilateral instead of same breast.

o   How long is the follow-up for the patients? Is it all 12 years as indicated in Figure 2.  

o   The cases that upgraded... What did they upgrade to? All DCIS? What was the nuclear grade? Was it extensive of focal? Were there calcifications?

o   Of the events noted in the DFS, lines 164-170, what were the events? DCIS? Invasive?

-        The discussion has good information but is confusing to read and should be reworked.

Comments on the Quality of English Language

The article would strongly benefit from an thorough English-language review as some of it is confusing and some difficult to read. The content is there but needs to be reworded.

Author Response

In the article “Unveiling Challenges in High-risk Breast Lesions: Atypical Ductal Hyperplasia and Lobular in situ Neoplasm” the authors set out to evaluate upgrade rates of the low-grade atypical in situ proliferations detected on vacuum assisted breast biopsies. Overall, I think this article does have interesting data but it needs to be further discussed/explained and some key data is missing. The article would benefit from an English Language review as much of it is confusing or structured in an odd way.

  1. The title doesn’t accurately reflect the content of the article. This article is focused on vacuum assisted breast biopsies and their subsequent upgrade rate. Also, the title doesn’t indicate what the challenge is. Unveiling is a bold term to use.

Thanks for the suggestion

We have changed the title to your suggestion and the academic editor's suggestion.

  1. Throughout the article the terms “upstage” or “upstaging” is used when I think it should all be “upgrade.” However, I speak American English so this may be used in other English-speaking countries.

Thank you for the suggestion.

We have changed the terms according to your suggestion

  1. As someone from the US who practices breast pathology, I am not as familiar with the UK B-coding system or the DIN and LIN categories. It may be worth expanding on these for those who may not be as familiar. Or include those categories with ones that others may be familiar with as I think it

Thank you we have included both classifications in the text (At the beginning of the text we have placed the second type of classification in brackets)

  1. Materials and Methods. How were the cases retrospectively identified? By the pathology or imaging? What timeframe were they collected? Were they consecutive cases? Further expand on the details of the collection.

we better specified the mode and time of patient selection.

  1.  

o   ADH and ALH/LCIS are often seen together in specimens. Were there no biopsies with both or were they excluded?

Thank you for the request. mixed forms were excluded

We have better specified the concept in the text.

  1. I assume these were all women but it is not explicitly stated.

yes, the patients were all women.

We made the concept explicit in the text

  1. Line 142. What do you mean by size? The size of the atypia? As noted, linear extent is often used to distinguish ADH from low-grade DCIS. If you have a 15 mm focus of ADH then one would likely call that DCIS. Or was this dimension the imaging target?

      I thank you for the question. The dimension of radiological imaging is intended in the text.  We specified    that in the text. 

  1. At our institution vacuum assist is most commonly used in cases of calcifications. In this study not all cases had calcifications. What was the reason for vacuum assist? And were all of these calcifications identified on specimen radiograph, the histology, or both?

The cases that did not present with microcalcifications presented as parenchymal distortions. In our institution, we plan to use vacuum-assisted biopsy in such cases. we have better specified this concept in the text

  1. What were the calcifications associated with in the histologic sections?

Following the biopsy, an x-ray of the cores is always taken to separate the cores with micros from the cores without microcalcifications

  1. Gauge of the needle was recorded as noted in the methods. An analysis evaluating needle gauge and upgrade rate would be good to include.

We thank you for your request. However, in our series, most patients (82.5%) underwent a biopsy with an 11 G needle. This makes a differentiated analysis by needle type less feasible from a statistical point of view. We reported the percentage of patients who underwent 11G needle biopsy in the results.

  1. Table 1. What is MMT? It isn’t defined anywhere. Include the definition of this abbreviation in the caption.

Thanks, MMT stands for mammotome (Vacuum assisted Breast Biopsy) but we agree it can be confusing. We changed the word to VABB.

  1. “Disease only specimen with micro” is confusing and I’m not sure what you mean by this. Based on the text I assume this is referring to the presence or absence of calcifications. Please reword. I would also note in the text that some cases had missing data about calcifications.

Thank you. We have explained better the materials and methods and, in the results, what we mean by disease only in cores with micro.

We also better specified when the lesions appeared in the form of microcalcifications.

  1. I think the formatting of Table 2 has been altered. I don’t know how to read it the way it currently is.

Thanks, sorry, we reformatted the table

  1. What is Table 3 demonstrating? There is no great explanation in the text.

Table 3 shows how upstaging of carcinoma in situ is more frequent in our series for ADH than for LIN 1 and 2 (p<0.001). There are no statistically significant differences in progression to invasive carcinoma between ADH and LIN1 and 2.

  1. Lines 168-169. Use ipsilateral instead of same breast.

Done

  1. How long is the follow-up for the patients? Is it all 12 years as indicated in Figure 2.  

The median time to follow up is 6.3 years (Q1-Q3: 2.5-10.3). we have added this data in the text.

  1. The cases that upgraded... What did they upgrade to? All DCIS? What was the nuclear grade? Was it extensive of focal? Were there calcifications?

From our overall series, we obtained 43 cases of upgrade to surgery to carcinoma in situ and 27 cases to invasive carcinoma. As regards the underestimations of carcinoma in situ, in 34/43 cases, there was an underestimation to low-grade ductal carcinoma in situ and in 9/43 cases an underestimation to intermediate grade ductal carcinoma in situ.​ We found an easier underestimation of the biopsy when the disease was not confined to the samples with microcalcifications. In all cases, the underestimated disease was focal and non-extensive.

Among the 27 patients with invasive carcinoma, those were the results:

- 12 Invasive ductal Carcinoma

- 2 Mucinous invasive Carcinoma

- 3 Tubular invasive Carcinoma

- 1 Cribriform invasive carcinoma

- 3 Mixed carcinoma, ductal and lobular

- 6 Lobular invasive carcinoma

We have specified this part better in the text

o Of the events noted in the DFS, lines 164-170, what were the events? DCIS? Invasive?

For ADH, we had 12 events during follow-up. five carcinomas in situ in the contralateral breast, four infiltrating carcinomas in the contralateral breast, three infiltrating carcinomas in the same breast.

For LIN1, we had ten events at follow-up: six ipsilateral infiltrating carcinomas, three ipsilateral carcinomas in situ, and one contralateral infiltrating carcinoma.

For Lin 2, we had six ipsilateral invasives, three ipsilateral in situ and one contralateral invasive.

We specified in the text.

The discussion has good information but is confusing to read and should be reworked.

we modified the discussion, ordering it, also according to the suggestions of the native English speaker

Comments on the Quality of English Language

The article would strongly benefit from an thorough English-language review as some of it is confusing and some difficult to read. The content is there but needs to be reworded.

Thank you, English has been reviewed by a native speaker.

Reviewer 2 Report

Comments and Suggestions for Authors

The authors describe their single institution experience with the evaluation of benign high risk lesions and advocate for the use of vacuum assisted breast biopsy and  excision to improve diagnosis and management of atypical ductal hyperplasia and LIN 1  and LIN 2.

Over the past decade, recommendations  of chemoprevention to treat these diseases has become standard of care.  How many of the 176 patients without upstaging at time of surgery were offered chemoprevention and followed through with treatment? Is there any data on this sub-population of patients and is there DFS different?

The formatting of Table 2 needs to be repaired

Comments on the Quality of English Language

The title reads unveiling challenges to high risk breast lesions and uses the terminology atypical ductal hyperplasia in the title and Lobular in situ Neoplasm

The contents of the manuscript describe LIN 1 and LIN2

Perhaps a better title would be Unveiling Challenges in High Risk Breast Lesions: Atypical ductal hyperplasia and Lobular Neoplasia.

The title of Table 2 and Table three should have an ampersand  between LIN1 and LIN2

Author Response

The authors describe their single institution experience with the evaluation of benign high-risk lesions and advocate for the use of vacuum assisted breast biopsy and excision to improve diagnosis and management of atypical ductal hyperplasia and LIN 1 and LIN 2.

Over the past decade, recommendations of chemoprevention to treat these diseases has become standard of care.  How many of the 176 patients without upstaging at time of surgery were offered chemoprevention and followed through with treatment? Is there any data on this sub-population of patients and is there DFS different?

In our case series 47/173 (27,1%), tamoxifen therapy was performed. The oncologist introduced this type of therapy and considered the various risk factors after a multidisciplinary discussion. The treatment was proposed to those patients with more critical risk factors, such as grade I familiarity with breast and ovarian neoplasm. We decided not to include this parameter in the analysis of DFS. DFS may, in fact, be influenced by other factors such as lifestyle, genetics, and breast density... which we would not be able to retrieve given the study's retrospective nature and which would also change the aim of the study itself to more focused on upstaging assessment. However, we introduced how many patients did therapy into the results and considered the non-inclusion in assessing DFS in the study limitations.

The formatting of Table 2 needs to be repaired

ok the table has been modified.

Comments on the Quality of English Language

The title reads unveiling challenges to high-risk breast lesions and uses the terminology atypical ductal hyperplasia in the title and Lobular in situ Neoplasm

The contents of the manuscript describe LIN 1 and LIN2

Perhaps a better title would be Unveiling Challenges in High-Risk Breast Lesions: Atypical ductal hyperplasia and Lobular Neoplasia.

Thank you we have changed the title in accordance with your suggestion and that of the other reviewers.

The title of Table 2 and Table three should have an ampersand between LIN1 and LIN2

Thanks, done

Reviewer 3 Report

Comments and Suggestions for Authors

This study is very interesting because all patients were operated after biopsy 

and the message about need for regular and prolonged monitoring 

is essential .

However , several point must be clarified and/or modified :

1/ The study period must be defined 

2/the BC family history is not quoted 

3/ the age groups must be specified in order to precise the long-term risk 

     <45/45-55/>55 or <50 and >50 

4/ the median follow-up of the study must be specified 

5/The median occurrence of B5a and B5b lesions should be detailed 

6/Table 2 is unclear ,please  express results in other form and clarify 

              the notion of "immediate upstaging" (  28.5%) and " subsequent upstaging " 

7/Did any patient take Tamoxifen ? 

8/Ref 5 ( major study ) is incomplete 

Author Response

This study is very interesting because all patients were operated after biopsy and the message about need for regular and prolonged monitoring is essential .

However , several point must be clarified and/or modified :

1/ The study period must be defined 

Thanks, done.

2/the BC family history is not quoted 

we specified familiarity

3/ the age groups must be specified in order to precise the long-term risk 

     <45/45-55/>55 or <50 and >50 

We divided the patients by age groups at biopsy and presented the data schematically in supplementary table one.

The difference in median age at biopsy was not statistically significant.

Variable

Level

ADH

(N=120)

LIN 1/2

(N=126)

P-value

Age at VABB, median (min-max)

52 (31-78)

49 (37-70)

0.056

Age at VABB, N (%)

<50

47 (39.2)

67 (53.2)

0.028

≥50

73 (60.8)

59 (46.8)

Age at VABB, N (%)

<45

24 (20.0)

26 (20.6)

0.025

4/ the median follow-up of the study must be specified 

The median time to follow up is 6.3 years (Q1-Q3: 2.5-10.3). we have added this data in the text

5/The median occurrence of B5a and B5b lesions should be detailed 

Thanks, done.

6/Table 2 is unclear, please express results in other form and clarify the notion of "immediate upstaging" (  28.5%) and " subsequent upstaging " thanks, the immediate upstage is the one found at surgery. the late upstage is the one identified at follow up.

Thanks, we changed the text according to your suggestion.

7/Did any patient take Tamoxifen? 

In our case series 47/173 (27,1%), tamoxifen therapy was performed. The oncologist introduced this type of therapy and considered the various risk factors after a multidisciplinary discussion. The treatment was proposed to those patients with more critical risk factors, such as grade I familiarity with breast and ovarian neoplasm. We decided not to include this parameter in the analysis of DFS. DFS may, in fact, be influenced by other factors such as lifestyle, genetics, and breast density... which we would not be able to retrieve given the study's retrospective nature and which would also change the aim of the study itself to more focused on upstaging assessment. However, we introduced how many patients did therapy into the results and considered the non-inclusion in assessing DFS in the study limitations.

8/Ref 5 ( major study ) is incomplete 

Thanks we changed the reference according to your suggestion

Reviewer 4 Report

Comments and Suggestions for Authors

The authors examined the upgrade rate of ADH and classic lobular neoplasia (ALH and classic LCIS) and assessed disease free survival (DFS) in those patients without an upgrade after surgical excision. They found high upgrade rates for both ADH and lobular neoplasia and no significant difference in DFS between the groups. They conclude that a multidisciplinary approach is best when determining further management. While the premise of the study is interesting, though not novel, the study design and analysis/presentation of results are flawed. There are several items the authors could address:

1.      It is unclear what exactly constitutes an upgrade in this study. The abstract (line 38) and methods (line 110) state the upgrade was defined as finding in situ and/or invasive carcinoma in the excision specimen. Does this include only ductal carcinoma in situ or non-classic types of lobular carcinoma in situ (i.e. pleomorphic and florid LCIS) as well? This should be clearly stated. Also line 42 of the abstract states upgrade to in situ carcinoma only – should be clarified/corrected.

2.      The introduction cites an upgrade rate of LIN (assuming LIN1/2 as included in study though not explicitly stated) of up to 67% (line 72). This seems extremely high as most will show a much lower rate unless non-classic types or rad-path discordant cases are included in upgrade rate results. Should double check this so as not to mislead readers.

3.      The methods describes a review of reported findings only and does not include a central pathology review to confirm the diagnosis of ADH/LIN. It would at least be good to explain whether cases are reviewed by subspecialty trained breast pathologists or general pathologists so the reader knows the level of expertise used in the evaluation of the biopsy specimens. Further, the distinction between ADH and low grade DCIS, as stated in the discussion, is based on qualitative and quantitative criteria which were created based on excisional biopsy findings, making precise distinction in small core biopsy samples difficult. The diagnosis of ADH has wide interobserver variability. A review of histology would be appropriate as some cases may be either considered borderline lesions or flat out DCIS by some pathologists and should then be excluded from this analysis or analyzed separately. A clear example of these differences in diagnostic interpretation is provided in Figure 1 – Figure 1 c shows (inexplicably) a huge amount of benign tissue and only a small corner of the atypical proliferation, which even at this cropped view looks more like DCIS than ADH. The stained slides (d and e) show more proliferative ducts (also mostly cropped) but again looks like clear DCIS (or the very at least ADH bordering on DCIS). The figure could be improved and this reviewer would suggest eliminating this case from analysis. 

4.      Continuing on suggestion above, there is no mention of an examination of the correlation between radiology and the pathology findings. Was rad-path correlation done? A 20% upgrade rate for classic lobular neoplasia would suggest one was not done and that several discordant cases were included (and possibly upgraded, arbitrarily elevating the upgrade rate as these may not actually be upgrades but rather sampling errors). A re-review of radiology and correlation with pathologic findings would be highly recommended.

5.      Did any patients have both ADH and LIN in their biopsies? If not, this should be stated.

6.      In Table 2, the rows are misaligned making it difficult to read. Table 3 is hard to interpret, no idea what is meant to be displayed here.

7.      In lines 168 and 169 there is a mention of “primitives”. I am not familiar with this terminology. What does that mean? It’s either ipsilateral (same or different quadrant) or contralateral. Primitive is not a term I’ve ever seen in the literature before.

8.      The discussion that ADH and LIN should be considered high risk lesions as a conclusion of this study is redundant. They have already been defined that way (in fact by the very authors themselves in the first line of the introduction). Not sure the extensive discussion of the need for more screening in high risk women is necessary as its standard of care in most institutions. 

Author Response

The authors examined the upgrade rate of ADH and classic lobular neoplasia (ALH and classic LCIS) and assessed disease free survival (DFS) in those patients without an upgrade after surgical excision. They found high upgrade rates for both ADH and lobular neoplasia and no significant difference in DFS between the groups. They conclude that a multidisciplinary approach is best when determining further management. While the premise of the study is interesting, though not novel, the study design and analysis/presentation of results are flawed. There are several items the authors could address:

 We thank you; we know that the topic is not new. However, the strength of our study is determined by a large number of patients for a single centre.

  1. It is unclear what exactly constitutes an upgrade in this study. The abstract (line 38) and methods (line 110) state the upgrade was defined as finding in situ and/or invasive carcinoma in the excision specimen.

we have precisely defined the upgrade in this way: the finding of an invasive or in situ carcinoma during surgery.

Does this include only ductal carcinoma in situ or non-classic types of lobular carcinoma in situ (i.e. pleomorphic and florid LCIS) as well?

in our series we found underestimates only in ductal carcinomas in situ. we have better specified the type of underestimation in the results.

This should be clearly stated.

Also line 42 of the abstract states upgrade to in situ carcinoma only – should be clarified/corrected.

We have better specified the concepts in the abstract. In particular, we found a difference between the upgrade between ADH and LIN in the upgrade to carcinoma in situ and no difference in the upgrade to infiltrating carcinoma between the two groups.

  1. The introduction cites an upgrade rate of LIN (assuming LIN1/2 as included in study though not explicitly stated) of up to 67% (line 72). This seems extremely high as most will show a much lower rate unless non-classic types or rad-path discordant cases are included in upgrade rate results. Should double check this so as not to mislead readers.

You're right, sorry. We revised the percentages according to the literature.

  1. The methods describes a review of reported findings only and does not include a central pathology review to confirm the diagnosis of ADH/LIN. It would at least be good to explain whether cases are reviewed by subspecialty trained breast pathologists or general pathologists so the reader knows the level of expertise used in the evaluation of the biopsy specimens.

all pathologists are dedicated breast pathologists with a high level of experience. The hospital is a third level oncology reference center for breast pathologies.

  1. Further, the distinction between ADH and low grade DCIS, as stated in the discussion, is based on qualitative and quantitative criteria which were created based on excisional biopsy findings, making precise distinction in small core biopsy samples difficult. The diagnosis of ADH has wide interobserver variability. A review of histology would be appropriate as some cases may be either considered borderline lesions or flat out DCIS by some pathologists and should then be excluded from this analysis or analyzed separately. A clear example of these differences in diagnostic interpretation is provided in Figure 1 – Figure 1 c shows (inexplicably) a huge amount of benign tissue and only a small corner of the atypical proliferation, which even at this cropped view looks more like DCIS than ADH. The stained slides (d and e) show more proliferative ducts (also mostly cropped) but again looks like clear DCIS (or the very at least ADH bordering on DCIS). The figure could be improved and this reviewer would suggest eliminating this case from analysis. 

In our opinion and according to ICD11 codyng the architectural characteristics of ADH are similar to those of low grade ductal carcinoma in situ.

The figure therefore seems completely correct to us and we think it is not correct to change it.

  1. Continuing on suggestion above, there is no mention of an examination of the correlation between radiology and the pathology findings. Was rad-path correlation done? A 20% upgrade rate for classic lobular neoplasia would suggest one was not done and that several discordant cases were included (and possibly upgraded, arbitrarily elevating the upgrade rate as these may not actually be upgrades but rather sampling errors). A re-review of radiology and correlation with pathologic findings would be highly recommended.

the correlation between radiology and pathology is always made in our institute and our upstaging percentage is perfectly in line with that of the literature.

Look for example:

DOI: 10.2214/AJR.11.7212

                 DOI: 10.1038/modpathol.2016.127

  1. Did any patients have both ADH and LIN in their biopsies? If not, this should be stated.

 we excluded mixed forms from our analysis. We have made this explicit in the text.

  1. In Table 2, the rows are misaligned making it difficult to read. Table 3 is hard to interpret, no idea what is meant to be displayed here.

we have changed, modified and better explained table 3.

  1. In lines 168 and 169 there is a mention of “primitives”. I am not familiar with this terminology. What does that mean? It’s either ipsilateral (same or different quadrant) or contralateral. Primitive is not a term I’ve ever seen in the literature before.

 Sorry, we changed the term according to your suggestion.

  1. The discussion that ADH and LIN should be considered high risk lesions as a conclusion of this study is redundant. They have already been defined that way (in fact by the very authors themselves in the first line of the introduction). Not sure the extensive discussion of the need for more screening in high-risk women is necessary as its standard of care in most institutions. 

We have modified the discussion trying to accommodate your requests. However, we think that in many institutions the monitoring of these patients is not considered "high risk"

We therefore believe that it is necessary to raise awareness of the need for careful follow-up of these patients.

Round 2

Reviewer 1 Report

Comments and Suggestions for Authors

The paper is well-written and the research well presented. I appreciate the authors' edits and thoughtful responses.  There are only a few minor changes I would suggest. 

- Table 1 -- change upstage to upgrade. 

- Line 191-192 -- I don't think familiarity is the correct term. Do you mean first degree relative?

- Line 228 -- "small dyscohesive epithelial cells"

- Line 274 -- lesions and diagnosis are a little redundant. Would say either "lesions" or "diagnoses."

Author Response

The paper is well-written and the research well presented. I appreciate the authors' edits and thoughtful responses.  There are only a few minor changes I would suggest. 

- Table 1 -- change upstage to upgrade.

Thanks for your suggestion.  We believe referred to Figure 1, which we changed by replacing upstage with upgrade.

- Line 191-192 -- I don't think familiarity is the correct term. Do you mean first degree relative?

Thanks for your suggestion.  We have replaced familiarity with relatives.

- Line 228 -- "small dyscohesive epithelial cells"

Thanks for your suggestion.  We have added cells in the sentence.

- Line 274 -- lesions and diagnosis are a little redundant. Would say either "lesions" or "diagnoses."

Thanks for your suggestion.  We have erased diagnosis.

Reviewer 4 Report

Comments and Suggestions for Authors

The authors address only issues concerning proofreading and readability and many were addressed only in comments to me and not specifically clarified in text. My issues with the study design and analysis remain.

The comment regarding Figure 1 (now Figure 3) is not clear to me. The authors state that diagnoses were made based on the WHO guidelines. Of course, the architectural features of ADH and low grade  DCIS are exactly the same, the distinction is made based on extent of duct involvement. The authors certainly know that arbitrary size cutoffs provided in WHO (and based not on core biopsy findings) are general guidelines. Additionally, ADH shows low grade atypia only. More cytologic atypia, regardless of extent and architectural features would not qualify for a diagnosis of ADH. In this example (still not even close to showing the entire extent of the atypia), the nuclear atypia looks more than simply low grade and the extent of duct involvement (in the small amount of proliferation available to reader) looks extensive. Certainly, if it is clearly ADH then, as previously suggested, modification of the images to show what is actually there would have been preferred. Questions the accuracy of other pathologic interpretations and does not address the low interobserver reproducibility in the distinction between the two.

The authors provide a reference to support their upgrade rate of LIN. It does fall into the range but trends toward those studies with discordant cases. One of the references provided (Ibrahim) was a flawed study as they had no rad-path correlation and did not exclude cases of pleomorphic LCIS.  A prospective, multi-institutional trial (PMCID: PMC4984674 DOI: 10.1245/s10434-015-4922-4) showed an upgrade of classic lobular neoplasia in rad-path concordant biopsies as 1%. A finding mirrored in many prior and subsequent studies, a reason why classic lobular neoplasia is not routinely required to undergo excision. The authors could at least address this discrepancy.  

Author Response

The authors address only issues concerning proofreading and readability and many were addressed only in comments to me and not specifically clarified in text. My issues with the study design and analysis remain.

The comment regarding Figure 1 (now Figure 3) is not clear to me. The authors state that diagnoses were made based on the WHO guidelines. Of course, the architectural features of ADH and low grade  DCIS are exactly the same, the distinction is made based on extent of duct involvement. The authors certainly know that arbitrary size cutoffs provided in WHO (and based not on core biopsy findings) are general guidelines. Additionally, ADH shows low grade atypia only. More cytologic atypia, regardless of extent and architectural features would not qualify for a diagnosis of ADH. In this example (still not even close to showing the entire extent of the atypia), the nuclear atypia looks more than simply low grade and the extent of duct involvement (in the small amount of proliferation available to reader) looks extensive. Certainly, if it is clearly ADH then, as previously suggested, modification of the images to show what is actually there would have been preferred. Questions the accuracy of other pathologic interpretations and does not address the low interobserver reproducibility in the distinction between the two.

Thank you for your suggestion. We completely changed the image deciding to accept your comment.

The authors provide a reference to support their upgrade rate of LIN. It does fall into the range but trends toward those studies with discordant cases. One of the references provided (Ibrahim) was a flawed study as they had no rad-path correlation and did not exclude cases of pleomorphic LCIS.  A prospective, multi-institutional trial (PMCID: PMC4984674 DOI: 10.1245/s10434-015-4922-4) showed an upgrade of classic lobular neoplasia in rad-path concordant biopsies as 1%. A finding mirrored in many prior and subsequent studies, a reason why classic lobular neoplasia is not routinely required to undergo excision. The authors could at least address this discrepancy.

Thank you for your suggestion. Our LIN upgrade rate is in agreement with the literature data reported in the Third International Consensus, published in 2023. “An upgrade into DCIS or invasive cancer is observed on an average of 20% of cases, with a wide range from 4 to 67% within the current literature”

  • Breast Tumours WHO Classification of Tumours, 5th Edition2019. https://nottingham repository.worktribe.com/output/4758580. Accessed 15 Jan 2023;
  • Elfgen C, Tausch C, Rodewald AK, Guth U, Rageth C, Bjelic-Radisic V et al (2022) Factors indicating surgical excision in classical type of lobular neoplasia of the breast. Breast Care (Basel) 17(2):121–128

In addition, in most of our cases there was a strong pathological-radiological discordance. Although a VABB was performed, most of the microcalcifications were fine pleomorphic (BI-RADS 4B), fine linear or fine linear branching (BI-RADS 4C), with linear or segmental distribution, or were true architectural pseudo distortion areas. Again, the literature says “The greatest indicator of an upgrade into invasive cancer, however, is a radiological discrepancy, such as a spiculated mass in clinical imaging and a histopathologic diagnosis of LN in the biopsy specimen of the same lesion”

  • Breast Tumours WHO Classification of Tumours, 5th Edition2019. https://nottingham-repository.worktribe.com/output/4758580. Accessed 15 Jan 2023;
  • Lewin AA, Mercado CL (2020) Atypical ductal hyperplasia and lobular neoplasia: update and easing of guidelines. AJR Am J Roentgenol 214(2):265–275;
  • Elfgen C, Tausch C, Rodewald AK, Guth U, Rageth C, Bjelic-Radisic V et al (2022) Factors indicating surgical excision in classical type of lobular neoplasia of the breast. Breast Care (Basel) 17(2):121–128
  • Girardi V, Guaragni M, Ruzzenenti N, Palmieri F, Fogazzi G, Cozzi A et al (2021) B3 Lesions at vacuum-assisted breast biopsy under ultrasound or mammography guidance: a single-center experience on 3634 consecutive biopsies. Cancers (Basel) 13(21):5443
  • Strachan C, Horgan K, Millican-Slater RA, Shaaban AM, Sharma N (2016) Outcome of a new patient pathway for managing B3 breast lesions by vacuum-assisted biopsy: time to change current UK practice? J Clin Pathol 69(3):248–254
  • Holbrook AI, Hanley K, Jeffers C, Kang J, Cohen MA (2019) Triaging atypical lobular hyperplasia and lobular carcinoma in situ on percutaneous core biopsy to surgery or observation: assiduous radiologic-pathologic correlation works, quantitating extent of disease does not. Arch Pathol Lab Med 143(5):621–627